# Prevalence and correlates of *Helicobacter pylori* infection among under-five children, adolescent and non-pregnant women in Nepal: Further analysis of Nepal national micronutrient status survey 2016

Suresh Mehata[1]*, Kedar Raj Parajuli[2], Narayan Dutt Pant[3], Binod Rayamajhee[4,5], Uday Narayan Yadav[6,7,8,9], Ranju Kumari Mehta[10], Priya Jha[11], Neha Mehta[12], Meghnath Dhimal[13], Dipendra Raman Singh[14]

1 Ministry of Health and Population, Government of Nepal, Kathmandu, Nepal, 2 Nutrition Section, Family Welfare Division, Department of Health Services, Ministry of Health and Population, Kathmandu, Nepal, 3 Grande International Hospital, Kathmandu, Nepal, 4 School of Optometry and Vision Science, Faculty of Medicine and health Sciences, UNSW, Sydney, Australia, 5 Department of Infection and Immunology, Kathmandu Research Institute for Biological Sciences (KRIBS), Lalitpur, Nepal, 6 Centre for Primary Health Care and Equity, UNSW, Sydney, Australia, 7 School of population Health, UNSW, Sydney, Australia, 8 Centre for Research Policy and Implementation, Biratnagar, Nepal, 9 Department of Public Health, Torrens University, Sydney, Australia, 10 Little Buddha College of Health Sciences, Min Bhawan, Kathmandu, Nepal, 11 Nepal Health Professional Council, Kathmandu, Nepal, 12 Institute of Medicine, Tribhuvan University Teaching Hospital, Kathmandu, Nepal, 13 Nepal Health Research Council, Kathmandu, Nepal, 14 Department of Health Services, Ministry of Health and Population, Kathmandu, Nepal

* sureshmht@gmail.com

**Data Availability Statement:** All relevant data are within the paper.

## Abstract

Most of the *Helicobacter pylori* infections occur in developing countries. The risk factors for *H. pylori* infections are poverty, overcrowding, and unhygienic conditions, which are common problems in under-privileged countries such as Nepal. Despite having a high risk of *H. pylori* infections, no national level study has been conducted to assess prevalence and correlates of *H. pylori* infection in Nepal. Therefore, we hypothesized that micronutrients such as iron, vitamin B12 deficiency, socio-economic status, and nutritional status correlate with the prevalence of *H. pylori* infection in Nepal.

We studied prevalence and correlates of *H. pylori* infection among under-five children, adolescents aged 10–19 years and married non-pregnant women aged 20–49 years using data from the Nepal National Micronutrient Status Survey 2016 (NNMSS-2016). *H. pylori* infection was examined in stool of 6–59 months old children and 20–49 years old non-pregnant women whereas the rapid diagnostic kit using blood sample was used among adolescent boys and girls.

Prevalence of *H. pylori* infection was 18.2% among 6–59 months old children, 14% among adolescent boys and 16% among adolescent girls aged 10–19 years; and 40% among 20–49 years non-pregnant women. Poor socioeconomic status, crowding, and unhygienic condition were found to be positively associated with higher incidence of *H. pylori*

**Funding:** The authors received no funding for this work.

**Competing interests:** The authors have declared that no competing interests exist.

infections. No significant correlation was observed between nutritional and micronutrients status (iron or risk of folate deficiency) with *H. pylori* infection.

Findings from this study suggest that poverty-associated markers are primary contributors of *H. pylori* infections in Nepalese communities. To control acquisition and persistence of *H. pylori* infection in Nepal, we suggest improved management of safe drinking water and implementation of sanitation and hygiene programs, with a focus on those of lower socio-economic status.

## Author summary

*Helicobacter pylori* is associated with a wide spectrum of gastrointestinal diseases and is a common problem in tropical region where inter-human contact is the primary mode of disease transmission. Poor socio-economic status is a crucial fueling factor of *H. pylori* infection. In this study, the authors present data from Nepal national micronutrient status survey 2016 (NNMSS-2016) and investigated the risk factors associated with *H. pylori* infections among under-five years old children, 10–19 years old adolescent boys and girls, and 20–49 years non-pregnant women nationwide. Study findings corroborate that poverty-associated markers are the key driving factor of *H. pylori* infections in Nepal, which can have a manifold effect on nutrition and subsequent child growth retardation. Management of safe drinking water and implementation of sanitation and hygiene practices to decrease acquisition of *H. pylori* infection is a pressing need in rural parts of Nepal and among marginalized communities. The findings of this study highlight the varied prevalence of *H. pylori* by age group, gender, place of residence, ethnic group, and ecological regions of the country. Moreover, no significant correlation was observed between nutritional and micronutrients status with *H. pylori* infection.

## Introduction

*Helicobacter pylori*, a gram-negative spiral-shaped bacterium previously known as *Campylobacter pyloridis*, was first isolated and identified by Warren and Marshall in 1982[1]. It colonizes the gastric milieu of more than half of the global population and is associated with gastric diseases such as peptic ulcer, chronic gastritis, gastric cancer, and mucosa associated lymphoid tissue (MALT) lymphoma[2, 3]. In addition, *H. pylori* has also been associated with some non-gastrointestinal diseases such as pre-eclampsia, autoimmune thyroid diseases, acute coronary diseases, myocardial infarction, hepatic encephalopathy, prostatitis, and psoriasis[4–7]. *H. pylori* is responsible for iron deficiency anemia, thrombocytopenia, fetal malformation, and fetal growth retardation in pregnant women. Epidemiological studies suggest that anemia due to iron deficiency is correlated with *H. pylori* infection. Gastric acidity caused by *H. pylori* can impair iron and vitamin B12 absorption and leads to anemia[8]. Further, among infants and children, the infection has been associated with chronic diarrhea and malnutrition[9, 10]. However, *H. pylori* infection associated as a protective factor for the diseases such as asthma, osteoporosis, inflammatory bowel disease, and esophageal cancer[4, 11, 12].

*H. pylori* is ubiquitous and the infection caused by the bacterium is a substantial global health problem, affecting 50% of the global population[2]. With infection rates of 50% of people in developed countries and 80% of people in developing countries, *H. pylori* infection is more concentrated in resource-limited countries[2]. In a meta-analysis, the overall global

prevalence of *H. pylori* infection was 44.3%, 50.8% in developing countries and 34.7% in developed countries[13]. There was no significant difference between the *H. pylori* infection prevalence between males and females (46.3% versus 42.7%)[13]. Most studies have demonstrated an equal rate of *H. pylori* infection between males and females; however, one study reported male gender as a significant risk factor[14]. The prevalence varies by geographical area, race and ethnicity, while the prevalence among children ranges from 10% to >80%[11]. Reports have shown the prevalence of infection is generally higher in developing countries due to socio-economic status, poverty; and unhygienic and overcrowded conditions[1, 13, 15–18].

*H. pylori* infection is often acquired during childhood and later the risk of infection rapidly declines[19]. For example, in developing countries, the infection is acquired during age below five years in 70–90% cases and generally remains asymptomatic leading to long term clinical sequelae such as gastritis, peptic ulcer and stomach cancer, then the risk of infection declines rapidly in later age[2, 20, 21]. In developed countries, the prevalence of infection in children is lower, and the percentage of infected people increases with age, accounting for up to 60% of cases[20, 22, 23]. However, most of the infections in developed countries are also acquired during childhood[24]. Despite high socio-economic status and hygienic conditions in developed countries, certain racial and ethnic groups (Blacks and Hispanics in the USA) have a higher rate of *H. pylori* associated infections[24]. This may be attributed to genetic predispositions to infection, which are yet to be fully understood[24].

The prevalence of *H. pylori* related infections in Nepal varies from 16.3% to 70.5%[25] and the infection is highly associated with malnutrition, and diarrheal diseases, notably among children. In a hospital based cross-sectional study reported from Nepal, patients who live in crowded urban area with lesser frequency of foods, were found to be highly susceptible to acquire the *H. pylori* infection[26]. Therefore, we hypothesized that micronutrients such as iron, vitamin B12 deficiency, socio-economic status, and nutritional status correlates with the prevalence of *H. pylori* infection in Nepal.

The aim of this study was to assess prevalence and correlates (socio-demographic, nutritional status, and micronutrient status) of *H. pylori* infection in Nepal among under five children, 10–19 years old adolescents and 20–49 years old non-pregnant women, using nationally representative data.

## Methods

### Ethics statement

Ethical approval for this study was obtained from the the Ethical Review Board (ERB) of the Nepal Health Research Council (NHRC) (Reg. No.: 201/2015). Well informed written consent was ascertained from all study participants before they were included in the study. Parental consent was obtained for the participants of <18 years. Adolescents of below 18 years were asked to participate in the study at the time of survey in community, eligible and interested adolescents written parents' consent was ascertained during the study.

### Data sources

Data for this study was taken from the Nepal National Micronutrient Status Survey 2016 (NNMSS-2016), a national level survey on basic health and demographic indicators[27]. The detailed methodology used has been presented in the same report[27]. In brief, the survey used stratified multi-stage cluster sampling. The five development regions (Eastern, Central, Western, Mid-western, and Far western) and three ecological zones (Terai, Hill, and Mountain) were used as the basis of stratifications; a random sampling approach across the 15 strata was used in order to represent the nation. A total of 180 clusters or wards (30 clusters from the

Mountain zone; 75 each from the Hill and Terai zones) were selected using probability proportionate to size, as the primary sampling units. Subsequently, systematic random sampling was used to select 24 households from each cluster. Total 4,309 households were selected for interview and the response was 99.7%. Further, the response rate for 1,709 children aged 6–59 months included in the study was 98.9%. Likewise, 1,025 and 1,865, 10–19 years old adolescent boys and girls were interviewed, respectively, with a response rate of 98% each. A total of 2,144 non-pregnant women aged 20–49 years were also interviewed with a response rate of 99%.

## Sample collection

Blood collected at the time of interview, and stool samples collected later that day or the following morning were used for *H. pylori* detection. From, 20–49 years old non-pregnant women and 6–59 months old children 11ml of venous blood was collected by the trained phlebotomists, while 6 ml was collected from 10–19 years old adolescent boys and girls. For children butterfly needles were used to collect venous blood. Blood samples were collected in three vacutainers (two 3ml purple top with EDTA and one 5ml blue top) following the standard precautions[27]. Some parameters such as, anemia (using a HemoCue Hb 301 analyzer) and RBC folate deficiency was assessed in the field by a phlebotomist using red blood cell folate microbiological assay[28].

**Anthropometric measurements.** Anthropometric measurements were collected to study the nutritional status of the children, adolescents, and adult women. A standard height/length-measuring board (Short-Board) was used to measure recumbent length of <2 years old children and standing height of >2 years old children, adolescent boys and girls, and adult women. For weight measurement, an electronic SECA digital scale (UNICEF Electronic Scale/Uni scale) was used.

Height-for-age (HAZ), weight-for-age (WAZ), and weight-for-length/height (WHZ) *z* scores were calculated using the WHO standards. The z-score <-1.96SD for stunting, underweight, and wasting were classified as HAZ, WAZ, and WHZ, respectively.

## Variables

**H. pylori infection.** Among children and non-pregnant women *H. pylori* infections were detected by testing antigen in stool samples[29], while among adolescents 10–19 years, the QuickVue Rapid Test Kit (RTK) (Quidel Corporation, San Diego, CA 92121, USA) for examination of *H. pylori* specific antibody (IgG) in whole blood of adolescent boys and girls was used. The QuickVue *H. pylori* RTK detects IgG antibodies specific to *H. pylori* produced by individuals colonized or infected with the organism. Approximately, 50μL (2 hanging drops) of whole blood was added to the test cassette from the Purple Top Vacutainer. Results were recorded as positive, negative or invalid after 5 minutes of samples load and assays were performed following the manufacturer's instructions. The reported sensitivity and specificity of QuickVue kit with biopsy was 90% and 78% respectively[30]. QuickVue has not been previously neither used nor validated for Nepalese population thus, we did not establish any cut off value for the sensitivity and specificity of QuickVue in this study. The validation of QuickVue in the local population in future studies is anticipated. Fresh stool samples collected were stored in a cold box of <10˚C and transferred to the pathologist. Then the pathologist segregated 1 gram of stool to a second sterile cryovial which was later transferred to Siddhi Polyclinic Laboratory (one of the most advanced molecular laboratory of Nepal) in Kathmandu to test for *H. pylori* antigen using cold chain box. All the frozen stool samples were stored in Walter Reed/AFRIMS Research Unit Nepal (WARUN) until laboratory tests were performed for the investigation of *H. pylori* antigens. *H. pylori* antigen detection was performed using an

enzyme-linked immunoassay (ELISA) test kit on a Mago clinical analyzer (Erba Lachema, Czech Republic). Positive and negative controls were used in each analytical test; and the absorbance for positive control was at least 0.8 OD units while that for negative control was less than 0.09 OD units. The reported sensitivity and specificity were 96% and 83%, respectively[31].

**External quality control of the laboratory analysis.** All laboratories that analyzed this study samples participated in CDC external quality assurance (EQA) programs. The EQA program also includes QA for ferritin, RBC folate, etc. The precision for the measurement of ferritin was >90–95% with <0.5% bias. The precision and bias for folate measurement was >90% and <4.0% respectively.

**Ferritin.** Ferritin, a blood protein, is a WHO recommended iron status indicator in human body, and low serum ferritin level indicates iron deficiency (WHO, 2001). To assess iron status, serum ferritin level was measured in venous blood samples collected from study participants and the geometric mean ferritin was calculated.

## Data analysis

All analyses were performed using the software Stata 15 (StataCorp LLC, Texas, USA). Adjusted prevalence ratio (APR) was calculated using multiple Poisson regression, with all covariates (age, gender, education, ecological zone, place of residence, wealth status, caste/ethnicity, nutritional status, serum ferritin, risk of folate deficiency and BMI) included simultaneously in the model considering the cluster sampling design. The wealth status was calculated based on household assets recommended by Demographic and Health Survey such as: type of toilets, cooking fuels, source of water and energy, type of housing, number of living rooms, ownership of household assets, etc. Principal component analysis was used to calculate the wealth index and further wealth quintile was categorized using ranked technique to translate it to the ordinal scale. $P < 0.05$ was considered to be statistically significant.

## Results

### Prevalence and correlates of *H. pylori* infection among children aged 6–59 months

Overall, 18.2% of children aged 6–59 months were infected with *H. pylori*. Study findings revealed a higher prevalence of *H. pylori* infection among participants aged 4–5 years (26%), a lower prevalence observed among participants aged below one year (9%), and that the prevalence of *H. pylori* increases increased with age. By ethnicity, the highest prevalence was observed among Muslims (31%) followed by Dalits (25%), whereas the lowest prevalence was observed among Brahmins/Chhetris (15%). Similarly, the highest prevalence was observed among those who resides in urban areas compared to those in rural areas (25% versus 19%, respectively). Moreover, the highest prevalence of *H. pylori* was observed among poor households (23%) compared to non-poor (17%) and adolescents who were at risk of folate deficiency (29.2%) (**Table 1**).

Multivariate Poisson regression analysis was used to ascertain the prevalence ratio of *H. pylori* infections among children aged 6–59 months. In the adjusted model, the higher prevalence of *H. pylori* infection was observed in Muslims (aPR2.15;95%CI:1.20–3.84) and Dalits (aPR 1.63;95% CI:1.13–2.36) as compared to Brahmins/Chhettris. Compared to poorest quintile, the lower prevalence of *H. pylori* infection was observed among the richer quintile participants (aPR 0.55; 95%CI:0.35–0.87).

Younger children had a lower likelihood of *H. pylori* infection; the prevalence ratio of *H. pylori* infection was higher among preschool children aged over four years (aPR 2.97; 95%CI:

**Table 1. Prevalence and correlates of *H. pylori* among preschool children (6 to 59 months) by background characteristics, nutritional, and micronutrient status.**

| | Distribution (%) | H. pylori result of 6–59 months children | | Correlates of H. pylori | | | Total (N) |
|---|---|---|---|---|---|---|---|
| | | Positive (%) | Negative (%) | aPR | 95%CI | P | |
| Age group (years) | | | | | | | |
| <1 | 9.0 | 8.9 | 91.1 | 1 | | | 144 |
| 1–2 | 21.3 | 11.0 | 89.0 | 1.16 | 0.55–2.41 | 0.698 | 309 |
| 2–3 | 21.9 | 22.2 | 77.8 | 2.42 | 1.18–4.95 | 0.016 | 348 |
| 3–4 | 24.0 | 22.7 | 77.3 | 2.52 | 1.30–4.89 | 0.006 | 362 |
| 4–5 | 23.8 | 26.2 | 73.8 | 2.97 | 1.48–5.93 | 0.002 | 360 |
| Gender | | | | | | | |
| Male | 54.7 | 20.0 | 80.0 | 1 | | | 778 |
| Female | 45.3 | 19.4 | 80.6 | 0.96 | 0.74–1.23 | 0.725 | 745 |
| Caste/ethnicity | | | | | | | |
| Brahmin/Chhetri | 31.0 | 14.6 | 85.4 | 1 | | | 542 |
| Terai-Madhesi Other Castes | 14.1 | 19.5 | 80.5 | 1.37 | 0.72–2.58 | 0.338 | 116 |
| Dalits | 18.3 | 25.1 | 74.9 | 1.63 | 1.08–2.47 | 0.019 | 316 |
| Janajatis | 32.6 | 20.2 | 79.8 | 1.18 | 0.80–1.73 | 0.401 | 489 |
| Muslims | 3.9 | 31.0 | 68.0 | 2.18 | 1.24–3.84 | 0.007 | 48 |
| Ecological Region | | | | | | | |
| Hill | 42.9 | 18.1 | 81.9 | 1 | | | 644 |
| Mountain | 8.0 | 23.3 | 76.7 | 1.04 | 0.68–1.59 | 0.855 | 250 |
| Terai | 49.1 | 20.5 | 79.5 | 1.44 | 0.99–2.11 | 0.058 | 629 |
| Place of residence | | | | | | | |
| Urban | 12.4 | 24.7 | 75.3 | 1 | | | 202 |
| Rural | 87.6 | 19.0 | 81 | 0.67 | 0.44–1.03 | 0.065 | 1321 |
| Province | | | | | | | |
| Province 1 | 14.6 | 25.3 | 74.7 | 1 | | | 231 |
| Province 2 | 23.2 | 16.9 | 83.1 | 0.47 | 0.21–1.06 | 0.067 | 174 |
| Bagmati Province | 20.4 | 21.4 | 78.6 | 1.05 | 0.62–1.76 | 0.860 | 206 |
| Gandaki Province | 8.9 | 12.5 | 87.5 | 0.61 | 0.28–1.33 | 0.218 | 135 |
| Lumbini Province | 16.3 | 19.9 | 80.1 | 0.72 | 0.42–1.22 | 0.221 | 287 |
| Karnali Province | 6.0 | 24.7 | 75.3 | 1.05 | 0.59–1.88 | 0.858 | 137 |
| Sudoorpachim Province | 10.6 | 17.9 | 82.1 | 0.70 | 0.42–1.18 | 0.182 | 353 |
| Wealth status | | | | | | | |
| Poorest | 21.0 | 22.5 | 77.5 | 1 | | | 422 |
| Poorest | 20.0 | 24.3 | 75.7 | 1.06 | 0.76–1.47 | 0.732 | 321 |
| Middle | 19.4 | 18.1 | 81.9 | 0.81 | 0.55–1.18 | 0.272 | 266 |
| Richer | 20.2 | 14.4 | 85.6 | 0.55 | 0.35–0.87 | 0.011 | 279 |
| Richest | 19.4 | 19.0 | 81.0 | 0.71 | 0.41–1.23 | 0.221 | 235 |
| Stunting (HAZ) | | | | | | | |
| No | 63.7 | 18.8 | 81.2 | 1 | | | 943 |
| Yes | 36.3 | 20.8 | 79.2 | 1.06 | 0.76–1.49 | 0.732 | 578 |
| Underweight (WAZ) | | | | | | | |
| No | 69.9 | 20.1 | 79.9 | 1 | | | 1062 |
| Yes | 30.1 | 18.2 | 81.8 | 0.73 | 0.49–1.10 | 0.138 | 459 |
| Wasting (WHZ) | | | | | | | |
| No | 88.0 | 19.8 | 80.2 | 1 | | | 1353 |
| Yes | 12.0 | 17.0 | 83.0 | 1.02 | 0.59–1.76 | 0.931 | 164 |

*(Continued)*

**Table 1.** (Continued)

| | Distribution (%) | H. pylori result of 6–59 months children | | Correlates of H. pylori | | | Total (N) |
|---|---|---|---|---|---|---|---|
| | | Positive (%) | Negative (%) | aPR | 95%CI | P | |
| Iron deficiency[a] | | | | | | | |
| No | 77.1 | 20.4 | 79.6 | 1 | | | 1166 |
| Yes | 22.9 | 17.8 | 82.3 | 1.06 | 0.77–1.46 | 0.739 | 317 |
| Risk of folate deficiency[b] | | | | | | | |
| No | 94.2 | 19.3 | 80.7 | 1 | | | 1401 |
| Yes | 5.8 | 29.2 | 70.8 | 1.32 | 0.96–1.81 | 0.091 | 75 |
| **Total** | **100** | **19.6** | **80.4** | | | | **1523** |

Abbreviations: HAZ: Height-for-Age Z scores; WAZ Weight-for-Age Z scores; WHZ: Weight-for-Height Z scores

[a]Biomarker was regression-adjusted to a pooled country reference to adjust for inflammation, using CRP and AGP (ferritin) or AGP only. Iron deficiency defined as inflammation-adjusted serum ferritin <12 μg/L[32].

[b]Folate cutoff based on the risk of megaloblastic anemia defined as RBC folate <305.0 nmol/L[32].

1.48–5.93), 3–4 years (aPR: 2.52; 95%CI: 1.30–4.89) and for those between 2–3 years (aPR: 2.42; 95%CI:1.18–4.89), compared to those aged less than one year (Table 1).

No significant difference in prevalence ratio was observed by nutritional status (stunting, wasting and underweight), iron and folic acid deficiency status of children with *H. pylori* infection (Table 1).

## Prevalence and correlates of *H. pylori* infection among adolescent boys aged 10–19 years

Overall, 14% of adolescent boys and 16% of adolescent girls were infected with *H. pylori* in Nepal. The prevalence was highest among adolescents who had a higher level of education (boys 28% and girls 30%) and those aged 15–19 years (boys 18% and girls 20%). Also, the highest prevalence was observed among adolescent boys who resides in Sudoorpachim Province (20%) and in urban areas (19%), followed by those in Terai-Madhesi ethnic groups and those who reside in the mountains (17% each). Additionally, among adolescent girls, the highest prevalence was observed among Muslims (25%) followed by those who reside in Sudoorpachim Province (21%) and in the Terai (17%) (**Tables 2 and 3**).

Tables 2 and 3 present adjusted prevalence ratios of *H.pylori* infection among adolescent boys and girls in Nepal by background characteristics. In the adjusted model, among adolescent boys, compared to those who never attended school or had some primary level education, a higher prevalence was observed among those who had higher-level education (aPR: 3.13; 95%CI: 1.22–8.00). Compared to the urban dwellers, a lower prevalence was observed among those who reside in rural areas (aPR: 0.59; 95% CI: 0.36–0.96). No statistically significant differences in prevalence were observed by age group, caste/ethnicity, ecological regions, provinces, height for age, BMI, or iron deficiency status.

Among adolescent girls, a higher prevalence of *H. pylori* infection was observed among adolescents aged 15–19 years (aPR: 1.49; 95%CI: 1.08–2.07) compared to those aged 10–14 years. By caste/ethnicity, a higher prevalence was observed among Muslims (aPR: 1.75; 95% CI:1.03–2.97) compared to Brahmins/Chhetris. No statistically significant differences in prevalence were observed by ecological region, province, wealth status, height for age, BMI, iron deficiency status, or the risk of folate deficiency (**Table 3**).

**Table 2. Prevalence and correlates of *H. pylori* among adolescent boys by background characteristics, nutritional status, and micronutrient status.**

| | Distribution (%) | *H. pylori* results | | Correlates of H. pylori | | | Total (N) |
|---|---|---|---|---|---|---|---|
| | | Positive (%) | Negative (%) | aPR | 95%CI | *P* | |
| Age group (years) | | | | | | | |
| 10–14 | 58.3 | 10.5 | 89.5 | 1 | | | 599 |
| 15–19 | 41.7 | 17.8 | 82.2 | 1.30 | 0.88–1.93 | 0.192 | 424 |
| Education | | | | | | | |
| Never attended school/ Primary | 32.9 | 9.1 | 90.9 | 1 | | | 327 |
| Secondary | 61.9 | 14.8 | 85.2 | 1.65 | 0.97–2.80 | 0.064 | 647 |
| Higher | 5.2 | 27.8 | 72.2 | 3.13 | 1.22–8.00 | 0.017 | 49 |
| Caste/ethnicity | | | | | | | |
| Brahmin/Chhetri | 35.7 | 15.2 | 84.5 | 1 | | | 434 |
| Terai-Madhesi Other Castes | 13.1 | 17.4 | 82.6 | 1.15 | 0.56–2.38 | 0.702 | 70 |
| Dalits | 14.4 | 10.4 | 89.6 | 0.62 | 0.36–1.07 | 0.087 | 159 |
| Janajatis | 33.9 | 11.9 | 88.1 | 0.94 | 0.61–1.45 | 0.787 | 338 |
| Muslims | 2.9 | 12.2 | 87.8 | 0.83 | 0.27–2.59 | 0.750 | 22 |
| Ecological Region | | | | | | | |
| Hill | 41.8 | 11.4 | 88.6 | 1 | | | 434 |
| Mountain | 6.9 | 17.3 | 82.7 | 1.23 | 0.67–2.27 | 0.503 | 157 |
| Terai | 51.3 | 14.8 | 85.2 | 1.65 | 0.97–2.81 | 0.064 | 432 |
| Place of residence | | | | | | | |
| Urban | 13.8 | 18.6 | 81.4 | 1 | | | 142 |
| Rural | 86.2 | 12.8 | 87.2 | 0.59 | 0.36–0.96 | 0.034 | 881 |
| Province | | | | | | | |
| Province 1 | 16.4 | 11.7 | 88.3 | 1 | | | 179 |
| Province 2 | 22.7 | 13.3 | 86.7 | 0.85 | 0.43–1.65 | 0.625 | 106 |
| Bagmati Province | 17.0 | 13.0 | 87.0 | 1.10 | 0.50–2.43 | 0.813 | 132 |
| Gandaki Province | 9.7 | 6.1 | 93.9 | 0.64 | 0.26–1.55 | 0.320 | 102 |
| Lumbini Province | 18.8 | 15.9 | 84.1 | 1.17 | 0.65–2.08 | 0.603 | 205 |
| Karnali Province | 4.4 | 19.8 | 80.2 | 1.55 | 0.74–3.24 | 0.248 | 87 |
| Sudoorpachim Province | 11.0 | 18.0 | 82.0 | 1.31 | 0.70–2.49 | 0.400 | 212 |
| Wealth status | | | | | | | |
| Poorest | 18.6 | 16.4 | 83.6 | 1 | | | 251 |
| Poorer | 20.3 | 10.9 | 89.1 | 0.62 | 0.34–1.13 | 0.117 | 211 |
| Middle | 22.2 | 12.4 | 87.6 | 0.51 | 0.28–0.92 | 0.025 | 209 |
| Richer | 17.5 | 14.5 | 85.5 | 0.60 | 0.32–1.14 | 0.121 | 165 |
| Richest | 21.5 | 14.1 | 85.9 | 0.53 | 0.26–1.08 | 0.079 | 187 |
| Stunting (HAZ) | | | | | | | |
| No | 66.6 | 12.2 | 87.8 | 1 | | | 636 |
| Yes | 33.4 | 15.5 | 84.5 | 1.37 | 0.96–1.95 | 0.086 | 343 |
| BMI (Wt/ht$^2$) | | | | | | | |
| <18.5 | 66.4 | 12.3 | 87.7 | 1 | | | 684 |
| 18.5–24.9 | 31.3 | 15.1 | 84.9 | 0.96 | 0.62–1.50 | 0.869 | 323 |
| ≥25.0 | 2.3 | 28.4 | 71.6 | 1.89 | 0.68–5.23 | 0.220 | 16 |
| Iron deficiency [a] | | | | | | | |
| No | 95.2 | 13.8 | 86.2 | 1 | | | 975 |
| Yes | 4.8 | 11.0 | 89.0 | 1.02 | 0.31–3.31 | 0.976 | 37 |

(*Continued*)

**Table 2.** (Continued)

| | Distribution (%) | H. pylori results | | Correlates of H. pylori | | | Total (N) |
|---|---|---|---|---|---|---|---|
| | | Positive (%) | Negative (%) | aPR | 95%CI | P | |
| Total | 100 | 14.0 | 86.0 | | | | 1023 |

Abbreviations: HAZ: Height-for-Age Z scores; BMI: Body Mass Index

[a]Biomarker was regression-adjusted to a pooled country reference to adjust for inflammation, using CRP and AGP (ferritin) or AGP only. Iron deficiency defined as inflammation-adjusted serum ferritin <15 µg/L[32].

[b]Folate cutoff based on the risk of megaloblastic anaemia defined as RBC folate <305.0 nmol/L[32].

### Prevalence and correlates of *H. pylori* infection among non-pregnant women aged 20–49 years

Overall, 40% of women of reproductive age had *H. pylori* infection. The study also revealed that the highest prevalence was observed among Muslims (61%), those who reside in Karnali Province (48%) and those who had never attended school or had some primary education (44%).

Table 4 shows the adjusted prevalence ratio of *H. pylori* among non-pregnant women in Nepal by socio-demographic status. In the adjusted model, a higher prevalence of *H. pylori* infection was observed among women belonging to ethnic groups: Indigenous/Janajatis (aPR: 1.25; 95%CI: 1.05–1.48) and Muslims (aPR: 1.56; 95%CI: 1.06–2.29) compared to Brahmins/ Chhetris. No statistically significant differences in prevalence were observed by age groups, education, ecological regions, place of residence, wealth status, height for age, BMI, the iron deficiency status, and the risk of folate deficiencies.

## Discussion

To our knowledge, this is the first study to include samples from all over Nepal including various age groups to gauge the prevalence of *H. pylori* colonization or infection among different age population of Nepal. In this study, we included children of <5 years, adolescents aged 10–19 years, and non-pregnant women aged 20–49 years. There are earlier studies on the prevalence of *H. pylori* infection in Nepal, though mostly in hospital settings with small sample size and not generalizable to the national population level[25, 26, 33].

Overall, 18% of pre-school children aged 6–59 months were infected with *H. pylori* with a higher prevalence of *H. pylori* infection among participants aged 4–5 years (26%). In a study from Vietnam, 39% of children up to 6 years were infected with *H. pylori*, but as in our study, they also reported a higher prevalence of infection with increased age[34]. A higher rate of *H. pylori* infection among children than that observed in our study was also reported in Brazil (70%)[35]. Moreover, the prevalence of *H. pylori* infection is lower in developed countries compared to developing countries[36, 37]. In this study, the prevalence of *H. pylori* infection was significantly correlated with socio-demographic factors such as ethnicity, place of residence, and wealth status of the study population. A higher rate of infection was found among children belonging to poor socioeconomic status those living in crowded spaces with poor hygienic conditions. In developing countries, the correlation between *H. pylori* infection and socioeconomic status is inconsistent. Some authors[38–40] have reported inverse correlations, a few have noted no relationship[41, 42] and one[43] has reported a significant correlation. The most common mode of transmission of *H. pylori* infection are the faeco-oral route and through contaminated water[14]. Additionally, cases of gastro-oral *H. pylori* transmission have been reported[44]. These modes of infection will have a higher likelihood of occurrence

**Table 3. Prevalence and correlates of *H. pylori* among adolescent girls by background characteristics, nutritional status, and micronutrient status.**

| | Distribution (%) | H. pylori results | | Correlates of H. pylori | | | Total (N) |
|---|---|---|---|---|---|---|---|
| | | Positive (%) | Negative (%) | aPR | 95%CI | P | |
| **Age group (years)** | | | | | | | |
| 10–14 | 55.3 | 12.5 | 87.5 | 1 | | | 994 |
| 15–19 | 44.7 | 20.1 | 79.9 | 1.49 | 1.08–2.07 | 0.016 | 817 |
| **Education** | | | | | | | |
| Never attended school/ Primary | 34.5 | 14.9 | 85.1 | 1 | | | 587 |
| Secondary | 61.6 | 15.6 | 84.4 | 0.99 | 0.73–1.36 | 0.973 | 1149 |
| Higher | 3.9 | 29.6 | 70.4 | 1.53 | 0.78–3.03 | 0.220 | 74 |
| **Caste/ethnicity** | | | | | | | |
| Brahmin/Chhetri | 33.2 | 13.8 | 86.2 | 1 | | | 694 |
| Terai-Madhesi Other Castes | 11.4 | 14.1 | 85.9 | 1.06 | 0.56–2.01 | 0.863 | 119 |
| Dalits | 16.4 | 22.8 | 77.2 | 1.45 | 0.92–2.27 | 0.107 | 314 |
| Janajatis | 36.7 | 14.8 | 85.2 | 1.09 | 0.77–1.56 | 0.623 | 648 |
| Muslims | 2.3 | 25.1 | 74.9 | 1.75 | 1.03–2.97 | 0.039 | 36 |
| **Ecological Region** | | | | | | | |
| Hill | 44.1 | 15.3 | 84.7 | 1 | | | 767 |
| Mountain | 7.5 | 12.8 | 87.2 | 0.73 | 0.49–1.08 | 0.118 | 284 |
| Terai | 48.4 | 17.0 | 83.0 | 1.33 | 0.89–1.99 | 0.168 | 760 |
| **Place of residence** | | | | | | | |
| Urban | 9.8 | 15.4 | 84.6 | 1 | | | 207 |
| Rural | 90.2 | 16.0 | 84.0 | 0.88 | 0.53–1.47 | 0.635 | 1604 |
| **Province** | | | | | | | |
| Province 1 | 16.4 | 15.4 | 84.6 | 1 | | | 291 |
| Province 2 | 21.1 | 15.2 | 84.8 | 0.85 | 0.45–1.59 | 0.602 | 196 |
| Bagmati Province | 17.1 | 13.8 | 86.2 | 1.02 | 0.60–1.73 | 0.937 | 205 |
| Gandaki Province | 10.4 | 9.0 | 91.0 | 0.71 | 0.43–1.17 | 0.181 | 184 |
| Lumbini Province | 18.2 | 18.4 | 81.6 | 1.01 | 0.62–1.67 | 0.954 | 365 |
| Karnali Province | 5.4 | 21.8 | 78.2 | 1.36 | 0.80–2.33 | 0.257 | 164 |
| Sudoorpachim Province | 11.5 | 20.6 | 79.4 | 1.39 | 0.85–2.27 | 0.190 | 406 |
| **Wealth status** | | | | | | | |
| Poorest | 23.0 | 19.2 | 80.8 | 1 | | | 486 |
| Poorer | 22.2 | 14.4 | 85.6 | 0.83 | 0.58–1.18 | 0.309 | 417 |
| Middle | 20.1 | 15.5 | 84.5 | 0.82 | 0.53–1.26 | 0.359 | 332 |
| Richer | 18.5 | 16.1 | 83.9 | 0.81 | 0.47–1.39 | 0.445 | 310 |
| Richest | 16.2 | 13.6 | 86.4 | 0.62 | 0.36–1.04 | 0.071 | 266 |
| **Stunting (HAZ)** | | | | | | | |
| No | 65.5 | 14.3 | 85.7 | 1 | | | 1097 |
| Yes | 34.5 | 18.2 | 81.8 | 1.20 | 0.93–1.54 | 0.163 | 594 |
| **BMI (Wt/ht$^2$)** | | | | | | | |
| <18.5 | 58.5 | 14.0 | 86.0 | 1 | | | 1038 |
| 18.5–24.9 | 39.2 | 18.5 | 81.5 | 1.12 | 0.82–1.53 | 0.485 | 730 |
| ≥25.0 | 2.3 | 21.3 | 78.7 | 1.54 | 0.73–3.25 | 0.255 | 40 |
| **Iron deficiency [a]** | | | | | | | |
| No | 82.2 | 15.3 | 84.7 | 1 | | | 1487 |
| Yes | 17.8 | 18.9 | 81.1 | 1.20 | 0.89–1.53 | 0.227 | 322 |
| **Risk of folate deficiency[b]** | | | | | | | |
| Yes | 16.2 | 14.3 | 85.7 | 1 | | | 239 |

(*Continued*)

**Table 3.** (*Continued*)

| | Distribution (%) | H. pylori results | | Correlates of H. pylori | | | Total (N) |
|---|---|---|---|---|---|---|---|
| | | Positive (%) | Negative (%) | aPR | 95%CI | P | |
| No | 83.8 | 16.3 | 83.7 | 0.74 | 0.53–1.02 | 0.066 | 1481 |
| **Total** | **100** | **15.9** | **84.1** | | | | **1811** |

Abbreviations: HAZ: Height-for-Age; BMIZ: BMI-Body Mass Index

[a]Biomarker was regression-adjusted to a pooled country reference to adjust for inflammation, using CRP and AGP (ferritin) or AGP only. Iron deficiency defined as inflammation-adjusted serum ferritin <15 μg/L[32].

[b]Folate cutoff based on the risk of megaloblastic anaemia defined as RBC folate <305.0 nmol/L[32].

among children of poor socioeconomic status, living in a crowded area and in unhygienic conditions. We postulate that women and children of lower socioeconomic status may have higher exposure to polluted water, such as that of ponds, hand-pumps, in the Terai region, which could explain the high prevalence of *H. pylori* infection among women and children of this area compared to other ecological regions in Nepal. A study by Bainganawt et al. (2014) supports the idea that the use of water from sources such as rivers and lakes increase the odds of *H. pylori* infection in pregnant women. Therefore, it is worth mentioning that improving water, sanitation, and hygiene practices may decrease the prevalence of *H. pylori* among children and women in Nepal.

Gender is considered as a potential risk factor of *H. pylori* infection[45]; thus, adolescent participants were segregated as boys and girls to compare the *H. pylori* incidence based on the gender. Among the adolescent population, 14% of boys and 16% of girls were positive for the anti-*H. pylori* IgG in rapid antibody tests performed by QuickVue kit. The prevalence was highest among adolescents who had a higher level of education (boys, 28%, and girls, 30%) and those aged 15–19 years (boys, 18%, and girls, 20%); highly educated people are more likely to be aware of hygiene practices and are therefore expected to have a lower infection rate of *H. pylori*. In the Nepalese context, adolescents aged 15–19 years who need to stay away from home in rented accommodation for their high school education may be exposed to external sources of *H. pylori*, including contaminated food from fast food retailers or street vendors with poor sanitation practices. Another reason could be the low socio-economic status of adolescents and their families which may force them to live in crowded and unhealthy environments where they have a higher likelihood of contact with *H. pylori*. Therefore, it is pertinent to consider socio-economic status and living/residential environments, lifestyle, and health behavior of an individual while designing interventions to control the *H. pylori* infections among adolescents.

Overall, 40% of non-pregnant women aged 20–49 years had an *H. pylori* infection, similar to results reported in Uganda (45.2%)[42]. The high rate of *H. pylori* infection was found among women belonging to marginalized communities, particularly among Dalits, Indigenous groups, and Muslims indicating that *H. pylori* poses a significant health concern among these groups in Nepal. This suggests that marginalized communities in the study population had poor access to clean food and clean water, unhygienic living conditions, and were of lower socioeconomic status. In the present study, we did not find an association between low birth weight and *H. pylori* infection, as demonstrated by a previous study reported in Uganda[46]. In this study, the prevalence of the infection varied according to the geographical location from 18% to 61%[42]. However, higher rates (56% to 74%) were reported from a U.S.–Mexico border population, and the prevalence again varied according to locality[47]. As in our study, Baingana et al. (2014) found an association between urban residence and education level and

**Table 4. Prevalence and correlates of *H. pylori* among Non-Pregnant Women by background characteristics, nutritional status, and micronutrient status.**

| | Distribution (%) | H. pylori | | Correlates of H. pylori | | | Total (N) |
|---|---|---|---|---|---|---|---|
| | | Positive (%) | Negative (%) | aPR | 95%CI | P | |
| Age group (Years) | | | | | | | |
| 20–24 | 23.3 | 38.6 | 61.4 | 1 | | | 395 |
| 25–34 | 40.0 | 40.8 | 59.2 | 1.04 | 0.85–1.27 | 0.732 | 702 |
| ≥35 | 36.7 | 41.1 | 58.9 | 1.03 | 0.83–1.27 | 0.804 | 640 |
| Education | | | | | | | |
| Never attended school/primary | 50.8 | 44.1 | 55.9 | 1 | | | 932 |
| Secondary | 33.8 | 37.7 | 62.3 | 0.86 | 0.70–1.06 | 0.162 | 544 |
| Higher | 15.4 | 34.3 | 65.7 | 0.80 | 0.61–1.05 | 0.103 | 243 |
| Caste/ethnicity | | | | | | | |
| Brahmin/Chhetri | 38.2 | 35.3 | 64.7 | 1 | | | 708 |
| Terai Madhesi Other Castes | 8.5 | 37.3 | 62.7 | 1.16 | 0.84–1.64 | 0.366 | 90 |
| Dalits | 14.5 | 43.7 | 56.3 | 1.21 | 0.94–1.55 | 0.149 | 270 |
| Janajatis | 36.9 | 44.1 | 55.9 | **1.25** | **1.05–1.48** | 0.013 | 637 |
| Muslims | 1.8 | 61.1 | 38.9 | **1.56** | **1.06–2.29** | 0.023 | 30 |
| Ecological Regions | | | | | | | |
| Hill | 42.7 | 38.1 | 61.9 | 1 | | | 746 |
| Mountain | 6.5 | 41.6 | 58.4 | 1.00 | 0.79–1.25 | 0.986 | 295 |
| Terai | 50.8 | 42.4 | 57.6 | 1.18 | 0.93–1.48 | 0.167 | 696 |
| Place of residence | | | | | | | |
| Urban | 13.8 | 42.1 | 57.9 | 1 | | | 248 |
| Rural | 86.2 | 40.1 | 59.9 | 0.90 | 0.74–1.09 | 0.300 | 1,489 |
| Province | | | | | | | |
| Province 1 | 17.2 | 47.7 | 52.3 | 1 | | | 294 |
| Province 2 | 19.5 | 34.3 | 65.7 | 0.63 | 0.42–0.97 | 0.036 | 171 |
| Bagmati Province | 22.4 | 37.7 | 62.3 | 0.85 | 0.64–1.14 | 0.273 | 247 |
| Gandaki Province | 10.8 | 31.1 | 68.9 | 0.73 | 0.47–1.13 | 0.160 | 211 |
| Lumbini Province | 16.2 | 42.2 | 57.8 | 0.84 | 0.60–1.13 | 0.240 | 321 |
| Karnali Province | 5.0 | 48.1 | 51.9 | 1.18 | 0.86–1.62 | 0.300 | 157 |
| Sudoorpachim Province | 9.0 | 47.9 | 52.1 | 1.02 | 0.78–1.32 | 0.911 | 336 |
| Wealth status | | | | | | | |
| Poorest | 14.9 | 44.8 | 55.2 | 1 | | | 376 |
| Poorer | 18.8 | 39.7 | 60.3 | 0.97 | 0.79–1.18 | 0.741 | 371 |
| Middle | 19.6 | 38.2 | 61.8 | 0.93 | 0.72–1.21 | 0.604 | 331 |
| Richer | 20.1 | 39.0 | 61.0 | 1.00 | 0.75–1.34 | 0.965 | 321 |
| Richest | 26.6 | 39.5 | 60.5 | 1.07 | 0.78–1.48 | 0.674 | 338 |
| Height | | | | | | | |
| ≥145 cm | 89.8 | 40.4 | 59.6 | 1 | | | 1,560 |
| <145 cm | 10.2 | 41.1 | 58.9 | 0.98 | 0.78–1.23 | 0.861 | 174 |
| BMI (Wt/ht$^2$) | | | | | | | |
| <18.5 | 12.8 | 40.8 | 59.2 | 1 | | | 232 |
| 18.5–24.9 | 61.4 | 41.2 | 58.8 | 1.00 | 0.82–1.22 | 0.986 | 1,093 |
| ≥25.0 | 25.8 | 38.5 | 61.5 | 0.93 | 0.73–1.17 | 0.516 | 409 |
| Iron deficiency[a] | | | | | | | |
| No | 81.7 | 41.4 | 58.6 | | | | 1439 |
| Yes | 18.3 | 33.5 | 66.5 | 0.90 | 0.74–1.09 | 0.280 | 291 |
| Risk of folate deficiency[b] | | | | | | | |

*(Continued)*

**Table 4.** (Continued)

| | Distribution (%) | *H. pylori* | | Correlates of *H. pylori* | | | Total (N) |
|---|---|---|---|---|---|---|---|
| | | Positive (%) | Negative (%) | aPR | 95%CI | P | |
| No | 90.2 | 40.1 | 59.9 | 1 | | | 1,453 |
| Yes | 9.8 | 42.5 | 57.5 | 1.04 | 0.82–1.30 | 0.756 | 191 |
| **Total** | **100** | **40.40** | **59.60** | | | | **1,737** |

Abbreviations: BMI: Body Mass Index

[a]Biomarker was regression-adjusted to a pooled country reference to adjust for inflammation, using CRP and AGP (ferritin) or AGP only. Iron deficiency defined as inflammation-adjusted serum ferritin <15 μg/L[32].

[b]Folate cutoff based on the risk of megaloblastic anemia defined as RBC folate <305.0 nmol/L[32].

the rate of *H. pylori* infection among pregnant women[42]. The rate of *H. pylori* infection largely depends upon geographic area, age, race, and socioeconomic status[1].

In our study, we did not find any significant correlation between iron deficiency status and risk of folate deficiency with *H. pylori* infection across the age groups, which was in accordance with the findings of other studies[48, 49]. However, some studies have reported association of *H. pylori* infection with iron deficiency, such that reduced serum ferritin levels were measured in *H. pylori* infected cases[50–52]. In addition, there are reports of reversal of chronic iron deficiency anemia through eradication of *H. pylori* infection, without any additional anemia targeted treatments[53]. Iron being essential factor for *H. pylori* growth, this bacterium may compete with host for iron uptake creating iron deficiency in hosts[54]. However, from analysis of different available literature, it can be concluded that there is no universal agreement on correlation of *H. pylori* infection with iron deficiency. Nonetheless, the discrepancy seen in the results from different studies may also be due to different sample size or different age group of patients used in different studies, as the correlation may be age specific.

## Strengths and limitations

This study is the largest nationwide epidemiological study to provide a comprehensive understanding of *H. Pylori* prevalence and correlated potential risk factors in Nepal. The study used systematic random sampling to estimate the prevalence, which enabled the recruitment of participants from diverse demographic backgrounds. This study has a few limitations. Because this study is cross-sectional, causality cannot be inferred from the findings. We were unable to account for several confounding factors, such as household sanitation status, family history and dietary habits, and thus residual confounding could not be eliminated. Future research should account for these factors. Furthermore, serology may create bias in differentiating past and present infection as antibody titers may remain high even after treatment. *H. pylori* positive rate was 26.2% in children at the age of 4–5 years, and 38.6% in non-pregnant women at the age of 20–24 years. However, the positive rates in adolescent boys and girls were very low at 10.5%-20.1%. While the *H. pylori* infection was evaluated by *H. pylori* antigen in the stool in children under five and non-pregnant women, it was evaluated only by rapid test kit in adolescent boys and girls. The differences in positive rate is probably due to the inadequate sensitivity and specificity of the both the test kits, the reported positive and negative predictive value of rapid test kit were 81% and 88%, respectively[31]. The positive and negative predictive values of ELISA test kit were 96% and 91%, respectively[32]. Hence, it is difficult to compare the data of adolescent boys and girls with the data obtained from subjects of other ages. Moreover, this study did not evaluate the health status of participants, such as chronic inflammatory diseases, can lead to decreased serum ferritin independent of *H. pylori* infection, and may have

impacted the results. Also, this study did not account for prior treatment with antibiotics or proton pump inhibitors. This study also did not assess the correlation of infection rate with symptoms and disease process. This study did not include adult men and no persons older than 49 years.

## Conclusion

In conclusion, the results of this study affirmed a sizeable prevalence of *H. pylori* infection in Nepal and was varied as per gender, age group, ethnic group, place of residence, and ecological region. We also reinforce the need for water, sanitation, and hygiene programs focused on marginalized communities and people of lower socio-economic status, aiming to improve hygiene practices, living conditions, and lifestyle behaviors that could decrease acquisition of *H. pylori* infection at both the household and community levels. This study do not find any correlation between nutritional and micronutrients status (iron and risk of RBC folate deficiencies) with *H. pylori* infection. Findings of this study suggest poverty-associated markers are the major drivers of *H. pylori* related infections in Nepalese communities. This study demonstrates the need for a population-based national study in order to understand the dynamics of *H. pylori* infection and transmission in Nepal.

## Acknowledgments

The authors would like to thank the team members from Ministry of Health and Population, New ERA, UNICEF, EU, USAID, and the CDC for conducting this research. We are very grateful to the Government of Nepal for providing the permission to use the data for analysis and publication of this work. Our sincere thank goes to Mr Todd Lewis for proof reading and editing the manuscript.

## Author Contributions

**Conceptualization:** Suresh Mehata, Kedar Raj Parajuli, Narayan Dutt Pant, Uday Narayan Yadav, Meghnath Dhimal, Dipendra Raman Singh.

**Data curation:** Suresh Mehata, Ranju Kumari Mehta.

**Formal analysis:** Suresh Mehata, Ranju Kumari Mehta.

**Methodology:** Suresh Mehata, Kedar Raj Parajuli, Uday Narayan Yadav.

**Project administration:** Suresh Mehata.

**Software:** Suresh Mehata.

**Writing – original draft:** Suresh Mehata, Narayan Dutt Pant, Binod Rayamajhee, Uday Narayan Yadav, Priya Jha, Neha Mehta.

**Writing – review & editing:** Suresh Mehata, Kedar Raj Parajuli, Narayan Dutt Pant, Binod Rayamajhee, Uday Narayan Yadav, Ranju Kumari Mehta, Priya Jha, Neha Mehta, Meghnath Dhimal, Dipendra Raman Singh.

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
