## [Decision Letter · Decision Letter 0]

3 Nov 2020

Dear Dr Mehata,

Thank you very much for submitting your manuscript "Prevalence and determinants of Helicobacter pylori infection among under-five children, adolescent and non-pregnant women in Nepal: A further analysis of Nepal national micronutrients status survey 2016" for consideration at PLOS Neglected Tropical Diseases. I sincerely apologize for the long delay in returning a decision on this manuscript. As with all papers reviewed by the journal, your manuscript was reviewed by members of the editorial board and by several independent reviewers. In light of the reviews (below this email), we would like to invite the resubmission of a significantly-revised version that takes into account the reviewers' comments. The reviewers were in agreement that many important details are missing from the current manuscript, as well as appropriate statistical analyses.

We cannot make any decision about publication until we have seen the revised manuscript and your response to the reviewers' comments. Your revised manuscript is also likely to be sent back to reviewers for further evaluation.

[1] A letter containing a detailed, point by point list of your responses to the review comments and a description of the changes you have made in the manuscript to address each concern. Please note while forming your response, if your article is accepted, you may have the opportunity to make the peer review history publicly available. The record will include editor decision letters (with reviews) and your responses to reviewer comments. If eligible, we will contact you to opt in or out.

Sincerely,

Jeffrey H Withey

Associate Editor

Robert Reiner

Deputy Editor

Reviewer's Responses to Questions

**Key Review Criteria Required for Acceptance?**

**Methods**

-Are the objectives of the study clearly articulated with a clear testable hypothesis stated?

-Is the study design appropriate to address the stated objectives?

-Is the population clearly described and appropriate for the hypothesis being tested?

-Is the sample size sufficient to ensure adequate power to address the hypothesis being tested?

-Were correct statistical analysis used to support conclusions?

-Are there concerns about ethical or regulatory requirements being met?

Reviewer #1: The lack of specific hypotheses makes the study way to exploratory and unfocused, despite the substantial data available regarding the burden of H pylori and the role of age, sex and poverty-related determinants

What rapid test kit was used, brand, model, date, etc. What was its Se and Sp, and impacts on results? Same for the stool antigen. Given that this is the main outcome of the study, it needs to be described in detail in the methods. And the impact of the differences between the two tests on the conclusion needs to be discussed also. How many samples were tested per plate and with repetitions or tittering levels?

What explains higher prevalence in adults when virtually no increasing trend is seen in younger individuals?

Data sources, describe age of adolescents and pregnant women, instead of reporting them in the sample collection section. 

Stata’s name is not “Software for Statistics and Data Science”, please provide the proper reference in standard format

Is age considered an effect modifier? What about the differences in tests? Prevalence in adolescents is substantially lower most likely due to the tests

Numerical covariates cannot just be adjusted as having a continuous linear relationship with the logodss of the H pylori prevalence without first verifying that this relationship is indeed linear.

Reviewer #2: - health status of the participants should be evaluated and clarified.

- H. pylori infection was evaluated only by rapid test kit in adolescent boys and girls. 

- the definition of the wealth status needs to be clarified in the text.

- the definition of "risk of folate deficiency" needs to be clarified in the text.

Reviewer #3: The tests for H. pylori (stool antigen test (monoclonal?) and serum/whole blood antibody test) are not named. This is needed information as it can influence the reliability of the results. It is not stated at all here what was tested in blood (likely antibodies, something like “to assess H. pylori infection antibodies against the pathogen were detected in whole blood using XX Test from YY company etc”). It is also important to know how many freeze and thaw cycles have been done to the stool samples. This might have a strong impact on the antigen test. It should be also clear whether the producer of the antigen test allows the testing of frozen stool samples. The temperatures of storage should also be stated.

Line 145-150: the response rate seems extremely high. Was this addressed in the cited reference? How is it explained?

Line 189: it should read probably IgG/IgM? It would sound better IgG and IgM

Line 188-190: it comes the question why those aged 10-19 were tested by blood test and not stool antigen test and vice versa.

Line 198-190: “Chi-square test was used to measure the association between the exploratory and outcome variables” This sentence is not clear. Chi-square can detect differences between groups but not measure the association, the formulation of the sentence is for a regression analysis. What were outcome variables? If I understand right there was only 1 (being infected or not).

**Results**

-Does the analysis presented match the analysis plan?

-Are the results clearly and completely presented?

-Are the figures (Tables, Images) of sufficient quality for clarity?

Reviewer #1: Why does table 1 lacks p-values or some test to compare groups? Also, given the sampling approach, bivariate results are probably not reasonable estimates. Bivariate tables can be integrated with regression tables, and avoid double description of the same results. There is too much redundancy in the results, the way the tables are presented

Too much emphasis on comparing boys with girls despite of not being a leading study hypothesis and no remarkable sex-related differences.

Cannot just say prevalence in differences parts of the paper due to using two very different tests

Reviewer #2: Results need to be reconsidered (see below).

Reviewer #3: Generally a lot of overly detailed results with minimal relevant information. Probably most tables could be provided as supplementary material. All results of an age group should be provided as one block. As there are no confidence intervals in the descriptive data tables it is very hard to see which differences are at all relevant.

Line 215-216: probably “highest” and “lowest” is meant here.

**Conclusions**

-Are the conclusions supported by the data presented?

-Are the limitations of analysis clearly described?

-Do the authors discuss how these data can be helpful to advance our understanding of the topic under study?

-Is public health relevance addressed?

Reviewer #1: Findings confirm that poverty is the main factor associated with the disease, as know from the literature. What is the novelty?

In the discussion, comparison of results need to take into account what tests were used for each study in each country. Even within this study, the changes in prevalence by age suggests to be test-dependent

Reviewer #2: Conclusion is clear.

Reviewer #3: Some very important points are not addressed at all in the discussion. There were no adult men and no persons older than 49 years in study, important points as they limit the statements that can be made about the prevalence of H. pylori infection in Nepal. 

Line 389: the sentence is formulated confusingly. In the other study children above 10 were highly infected? How high? This point should be addressed as the data of the study is also from Nepal and puts into question whether methodological problems may be the cause (e.g., how was H. pylori infection detected in the other study?, If it was stool antigen testing maybe it was from fresh samples?)

**Editorial and Data Presentation Modifications?**

Reviewer #1: The English writing requires some minor adjustments and typographical mistakes

Somewhat lengthy introduction with an excessive emphasis on developed countries in the introduction, which are not relevant for comparison to the settings of Nepal. I suggest focusing on more comparable settings to the study population

Reviewer #2: Editorial modification is required (see below).

Reviewer #3: Line 91-92: what is meant here by marginally? Is it then not relevant?

Line 92-93: “H pylori is cause of gastrointestinal blood loss, reduction in iron absorption and increase in iron absorption by bacteria.” This sentence does not have a reference. It is also too general with stating H. pylori is a cause of GI blood loss. Is it a relevant cause?

Line 95-97: these are more associations between H. pylori and the diseases mentioned then causing a clear decrease.

Line 101-104: Probably here the study from Hooi et al (Global Prevalence of Helicobacter pylori Infection: Systematic Review and Meta-Analysis) is mentioned. It would be better to cite the original study (the meta-analysis) then papers citing it. It also looks strange to cite 2 papers (14 and 15) for the same data.

Line 110-112: the sentence is not clear? Bloating and vomiting are risk factors for H. pylori infection? In what time period, compared to whom? Most person have bloating or vomiting sometime in their lives. Also family history of gastric cancer is probably not a risk factor for H. pylori infection but rather an association with (most likely the risk factor for cancer is H. pylori infection.

Line 113-117: In the first sentence it is not clear what is meant by longer period. The second sentence is repeating the first (acquired at an early age is during childhood). In developed countries time of infection for most persons is childhood. The whole sentence is rather confusing and too long, it should read probably something like “…acquired during childhood after which the risk of infection rapidly declines…”.

Some sentences in the manuscript are very long and not formulated clearly. For example in line 81-85 it reads as if the lymphoma was colonizing the stomach: “Helicobacter pylori (H. pylori), a gram-negative and spiral-shaped bacterium previously known as Campylobacter pyloridis then C. pylori, was first isolated and identified by Warren and Marshall in 1982 (1), has been associated with gastric diseases such as peptic ulcer, chronic gastritis, gastric cancer, mucosa associated lymphoid tissue (MALT) lymphoma (2, 3) which colonizes in the gastric milieu of more than half of the global population.”

Italicization and abbreviations are not consistent in the text.

**Summary and General Comments**

Reviewer #1: This is an interesting effort, but unfocused and without clear hypotheses relevant to the global community in a topic where substantial evidence already exists. The manuscript needs a clear direction and a central topic

Reviewer #2: In this manuscript, Mehata et al investigated the prevalence of H. pylori infection among the Nepalese population. They found that poverty is associated with high prevalence of H. pylori infection.

1) In the Abstract section (l.39), mucosa associated lymphoid tissue lymphomas are indolent in nature and are rarely a cause of death, so this sentence needs to be corrected.

2) In the Background section, the authors wrote that gastric acidity caused by H. pylori leads to reduced vitamin B12 (l.91-92), but this is not true. Hypochlorhydria which occurs as a result of atrophic gastritis caused by H. pylori infection leads to reduced vitamin B12. This misleading sentence needs to be corrected.

3) In the Background section, the reviewer could not understand the sentence "reduction in iron absorption and increase in iron absorption by bacteria" (l.93). This sentence is confusing and needs to be edited.

4) The paragraph in the Background section from line 98 to123 has conflicting sentences within itself and is not so important for the present study. The authors should consider deleting or carefully re-editing this paragraph.

5) Although the authors hypothesized that supply of micronutrients determines the prevalence of H. pylori infection in Nepal (l.127-128), there is no sentence in the Results or Discussion verifying this hypothesis. Furthermore, the only micronutrient assessed in this study, other than iron, is folate (described as "risk of folate deficiency"), which was not associated with H. pylori infection. The authors should investigate other additional micronutrients, such as vitamin B12, selenium, etc., if they wish to prove their hypothesis. In addition, the definition of "risk of folate deficiency" needs to be clarified in the text.

6) Were all the participants healthy? The authors are using serum ferritin as the parameter of iron status. Some diseases, such as chronic inflammatory diseases, can lead to decreased serum ferritin independent of H. pylori infection, so the health status of the participants should be evaluated and clarified.

7) H. pylori positive rate was 26.2% in children at the age of 4-5, and 38.57% in non-pregnant women at the age of 20-24. However, the positive rates in adolescent boys and girls were very low at 10.5%-20.1%. While the H. pylori infection was evaluated by H. pylori antigen in the stool in children under five and non-pregnant women, it was evaluated only by rapid test kit in adolescent boys and girls. This decreased positive rate is probably due to the inadequate sensitivity of the rapid test kit. Hence, it is difficult to compare the data of adolescent boys and girls with the data obtained from subjects of other ages. The authors need to explain this limitation in the text.

8) How did the authors judge the wealth status of the participants? This needs to be clarified in the text.

9) There are a number of errors in the Discussion section (e.g. l.370 the authors describe that H. pylori infection was significantly associated with residence area in children under the age of 5, but none of the provinces or place of residence were significantly associated with H. pylori infection./ l.405 Dalits is not significantly associated with H. pylori infection./ etc.). The authors need to carefully check the manuscript and correct the errors.

10) The reference needs to be edited to match the style of this journal.

11) There are many grammatical errors in the text. These need to be corrected.

Reviewer #3: The study tries to estimate the prevalence of H. pylori infected individuals in Nepal using data and samples already available from an earlier population survey.

Line 124-130: if the prevalence of infection in Nepal is already known then it should be stated more clearly what this study adds to already available data (the study aim states “study was to assess prevalence and determinants of H. pylori”). Different study population, micronutrient data?

On the same line the sentence "Hence, we hypothesized that supply of micronutrients determines the prevalence of H. pylori

infection in Nepal." is unclear as then it is not stated how this hypothesis was then tested.

PLOS authors have the option to publish the peer review history of their article (what does this mean?). If published, this will include your full peer review and any attached files.

Reviewer #1: No

Reviewer #2: No

Reviewer #3: No
---

## [Editor Report · Decision Letter 1]

5 Feb 2021

Dear Dr Mehata,

Thank you very much for submitting your manuscript "Prevalence and correlates of Helicobacter pylori infection among under-five children, adolescent and non-pregnant women in Nepal: Further analysis of Nepal national micronutrient status survey 2016" for consideration at PLOS Neglected Tropical Diseases. 

There was no response to reviewers' comments included with this revision- only a response to editorial comments. This paper cannot be evaluated further until you respond to each of the reviewers comments directly and submit a revised manuscript with this point by point response. These reviewer comments were included in the original decision letter. I have copied them at the bottom of this letter as well.

We cannot make any decision about publication until we have seen the revised manuscript and your response to the reviewers' comments. Your revised manuscript is also likely to be sent to reviewers for further evaluation.

Sincerely,

Jeffrey H Withey

Associate Editor

Robert Reiner

Deputy Editor

There is no response to reviewers comments included with this revision- only a response to editorial comments. This paper cannot be evaluated further until you respond to each of the reviewers comments directly and submit a revised manuscript with this point by point response.

Reviewer #1: The lack of specific hypotheses makes the study way to exploratory and unfocused, despite the substantial data available regarding the burden of H pylori and the role of age, sex and poverty-related determinants

What rapid test kit was used, brand, model, date, etc. What was its Se and Sp, and impacts on results? Same for the stool antigen. Given that this is the main outcome of the study, it needs to be described in detail in the methods. And the impact of the differences between the two tests on the conclusion needs to be discussed also. How many samples were tested per plate and with repetitions or tittering levels?

What explains higher prevalence in adults when virtually no increasing trend is seen in younger individuals?

Data sources, describe age of adolescents and pregnant women, instead of reporting them in the sample collection section.

Stata’s name is not “Software for Statistics and Data Science”, please provide the proper reference in standard format

Is age considered an effect modifier? What about the differences in tests? Prevalence in adolescents is substantially lower most likely due to the tests

Numerical covariates cannot just be adjusted as having a continuous linear relationship with the logodss of the H pylori prevalence without first verifying that this relationship is indeed linear.

Reviewer #2: - health status of the participants should be evaluated and clarified.

- H. pylori infection was evaluated only by rapid test kit in adolescent boys and girls.

- the definition of the wealth status needs to be clarified in the text.

- the definition of "risk of folate deficiency" needs to be clarified in the text.

Reviewer #3: The tests for H. pylori (stool antigen test (monoclonal?) and serum/whole blood antibody test) are not named. This is needed information as it can influence the reliability of the results. It is not stated at all here what was tested in blood (likely antibodies, something like “to assess H. pylori infection antibodies against the pathogen were detected in whole blood using XX Test from YY company etc”). It is also important to know how many freeze and thaw cycles have been done to the stool samples. This might have a strong impact on the antigen test. It should be also clear whether the producer of the antigen test allows the testing of frozen stool samples. The temperatures of storage should also be stated.

Line 145-150: the response rate seems extremely high. Was this addressed in the cited reference? How is it explained?

Line 189: it should read probably IgG/IgM? It would sound better IgG and IgM

Line 188-190: it comes the question why those aged 10-19 were tested by blood test and not stool antigen test and vice versa.

Line 198-190: “Chi-square test was used to measure the association between the exploratory and outcome variables” This sentence is not clear. Chi-square can detect differences between groups but not measure the association, the formulation of the sentence is for a regression analysis. What were outcome variables? If I understand right there was only 1 (being infected or not).

Results

-Does the analysis presented match the analysis plan?

-Are the results clearly and completely presented?

-Are the figures (Tables, Images) of sufficient quality for clarity?

Reviewer #1: Why does table 1 lacks p-values or some test to compare groups? Also, given the sampling approach, bivariate results are probably not reasonable estimates. Bivariate tables can be integrated with regression tables, and avoid double description of the same results. There is too much redundancy in the results, the way the tables are presented

Too much emphasis on comparing boys with girls despite of not being a leading study hypothesis and no remarkable sex-related differences.

Cannot just say prevalence in differences parts of the paper due to using two very different tests

Reviewer #2: Results need to be reconsidered (see below).

Reviewer #3: Generally a lot of overly detailed results with minimal relevant information. Probably most tables could be provided as supplementary material. All results of an age group should be provided as one block. As there are no confidence intervals in the descriptive data tables it is very hard to see which differences are at all relevant.

Line 215-216: probably “highest” and “lowest” is meant here.

Conclusions

-Are the conclusions supported by the data presented?

-Are the limitations of analysis clearly described?

-Do the authors discuss how these data can be helpful to advance our understanding of the topic under study?

-Is public health relevance addressed?

Reviewer #1: Findings confirm that poverty is the main factor associated with the disease, as know from the literature. What is the novelty?

In the discussion, comparison of results need to take into account what tests were used for each study in each country. Even within this study, the changes in prevalence by age suggests to be test-dependent

Reviewer #2: Conclusion is clear.

Reviewer #3: Some very important points are not addressed at all in the discussion. There were no adult men and no persons older than 49 years in study, important points as they limit the statements that can be made about the prevalence of H. pylori infection in Nepal.

Line 389: the sentence is formulated confusingly. In the other study children above 10 were highly infected? How high? This point should be addressed as the data of the study is also from Nepal and puts into question whether methodological problems may be the cause (e.g., how was H. pylori infection detected in the other study?, If it was stool antigen testing maybe it was from fresh samples?)

Editorial and Data Presentation Modifications?

Reviewer #1: The English writing requires some minor adjustments and typographical mistakes

Somewhat lengthy introduction with an excessive emphasis on developed countries in the introduction, which are not relevant for comparison to the settings of Nepal. I suggest focusing on more comparable settings to the study population

Reviewer #2: Editorial modification is required (see below).

Reviewer #3: Line 91-92: what is meant here by marginally? Is it then not relevant?

Line 92-93: “H pylori is cause of gastrointestinal blood loss, reduction in iron absorption and increase in iron absorption by bacteria.” This sentence does not have a reference. It is also too general with stating H. pylori is a cause of GI blood loss. Is it a relevant cause?

Line 95-97: these are more associations between H. pylori and the diseases mentioned then causing a clear decrease.

Line 101-104: Probably here the study from Hooi et al (Global Prevalence of Helicobacter pylori Infection: Systematic Review and Meta-Analysis) is mentioned. It would be better to cite the original study (the meta-analysis) then papers citing it. It also looks strange to cite 2 papers (14 and 15) for the same data.

Line 110-112: the sentence is not clear? Bloating and vomiting are risk factors for H. pylori infection? In what time period, compared to whom? Most person have bloating or vomiting sometime in their lives. Also family history of gastric cancer is probably not a risk factor for H. pylori infection but rather an association with (most likely the risk factor for cancer is H. pylori infection.

Line 113-117: In the first sentence it is not clear what is meant by longer period. The second sentence is repeating the first (acquired at an early age is during childhood). In developed countries time of infection for most persons is childhood. The whole sentence is rather confusing and too long, it should read probably something like “…acquired during childhood after which the risk of infection rapidly declines…”.

Some sentences in the manuscript are very long and not formulated clearly. For example in line 81-85 it reads as if the lymphoma was colonizing the stomach: “Helicobacter pylori (H. pylori), a gram-negative and spiral-shaped bacterium previously known as Campylobacter pyloridis then C. pylori, was first isolated and identified by Warren and Marshall in 1982 (1), has been associated with gastric diseases such as peptic ulcer, chronic gastritis, gastric cancer, mucosa associated lymphoid tissue (MALT) lymphoma (2, 3) which colonizes in the gastric milieu of more than half of the global population.”

Italicization and abbreviations are not consistent in the text.

Summary and General Comments

Reviewer #1: This is an interesting effort, but unfocused and without clear hypotheses relevant to the global community in a topic where substantial evidence already exists. The manuscript needs a clear direction and a central topic

Reviewer #2: In this manuscript, Mehata et al investigated the prevalence of H. pylori infection among the Nepalese population. They found that poverty is associated with high prevalence of H. pylori infection.

1) In the Abstract section (l.39), mucosa associated lymphoid tissue lymphomas are indolent in nature and are rarely a cause of death, so this sentence needs to be corrected.

2) In the Background section, the authors wrote that gastric acidity caused by H. pylori leads to reduced vitamin B12 (l.91-92), but this is not true. Hypochlorhydria which occurs as a result of atrophic gastritis caused by H. pylori infection leads to reduced vitamin B12. This misleading sentence needs to be corrected.

3) In the Background section, the reviewer could not understand the sentence "reduction in iron absorption and increase in iron absorption by bacteria" (l.93). This sentence is confusing and needs to be edited.

4) The paragraph in the Background section from line 98 to123 has conflicting sentences within itself and is not so important for the present study. The authors should consider deleting or carefully re-editing this paragraph.

5) Although the authors hypothesized that supply of micronutrients determines the prevalence of H. pylori infection in Nepal (l.127-128), there is no sentence in the Results or Discussion verifying this hypothesis. Furthermore, the only micronutrient assessed in this study, other than iron, is folate (described as "risk of folate deficiency"), which was not associated with H. pylori infection. The authors should investigate other additional micronutrients, such as vitamin B12, selenium, etc., if they wish to prove their hypothesis. In addition, the definition of "risk of folate deficiency" needs to be clarified in the text.

6) Were all the participants healthy? The authors are using serum ferritin as the parameter of iron status. Some diseases, such as chronic inflammatory diseases, can lead to decreased serum ferritin independent of H. pylori infection, so the health status of the participants should be evaluated and clarified.

7) H. pylori positive rate was 26.2% in children at the age of 4-5, and 38.57% in non-pregnant women at the age of 20-24. However, the positive rates in adolescent boys and girls were very low at 10.5%-20.1%. While the H. pylori infection was evaluated by H. pylori antigen in the stool in children under five and non-pregnant women, it was evaluated only by rapid test kit in adolescent boys and girls. This decreased positive rate is probably due to the inadequate sensitivity of the rapid test kit. Hence, it is difficult to compare the data of adolescent boys and girls with the data obtained from subjects of other ages. The authors need to explain this limitation in the text.

8) How did the authors judge the wealth status of the participants? This needs to be clarified in the text.

9) There are a number of errors in the Discussion section (e.g. l.370 the authors describe that H. pylori infection was significantly associated with residence area in children under the age of 5, but none of the provinces or place of residence were significantly associated with H. pylori infection./ l.405 Dalits is not significantly associated with H. pylori infection./ etc.). The authors need to carefully check the manuscript and correct the errors.

10) The reference needs to be edited to match the style of this journal.

11) There are many grammatical errors in the text. These need to be corrected.

Reviewer #3: The study tries to estimate the prevalence of H. pylori infected individuals in Nepal using data and samples already available from an earlier population survey.

Line 124-130: if the prevalence of infection in Nepal is already known then it should be stated more clearly what this study adds to already available data (the study aim states “study was to assess prevalence and determinants of H. pylori”). Different study population, micronutrient data?

On the same line the sentence "Hence, we hypothesized that supply of micronutrients determines the prevalence of H. pylori

infection in Nepal." is unclear as then it is not stated how this hypothesis was then tested.
---

## [Decision Letter · Decision Letter 2]

30 Mar 2021

Dear Dr Mehata,

Thank you very much for submitting your manuscript "Prevalence and correlates of Helicobacter pylori infection among under-five children, adolescent and non-pregnant women in Nepal: Further analysis of Nepal national micronutrient status survey 2016" for consideration at PLOS Neglected Tropical Diseases. As with all papers reviewed by the journal, your manuscript was reviewed by members of the editorial board and by several independent reviewers. The reviewers appreciated the attention to an important topic. Based on the reviews, we are likely to accept this manuscript for publication, providing that you modify the manuscript according to the review recommendations. 

One reviewer still has concerns with the first revisions of this manuscript and you must submit a detailed response to these concerns along with the second revision of the manuscript.

Sincerely,

Jeffrey H Withey

Associate Editor

Robert Reiner

Deputy Editor

Reviewer's Responses to Questions

**Key Review Criteria Required for Acceptance?**

**Methods**

-Are the objectives of the study clearly articulated with a clear testable hypothesis stated?

-Is the study design appropriate to address the stated objectives?

-Is the population clearly described and appropriate for the hypothesis being tested?

-Is the sample size sufficient to ensure adequate power to address the hypothesis being tested?

-Were correct statistical analysis used to support conclusions?

-Are there concerns about ethical or regulatory requirements being met?

Reviewer #1: The hypothesis posed in the introduction are related to nutrition-related covariates, but also include socioeconomic status, preventing a focused analysis and manuscript. This reflects in the introduction, results and section, that try to address any potential factor associated to the outcome, without focus or set direction. Similarly, the micronutrient-related hypotheses end up receiving little attention and are lost in the manuscript, particularly in the central messages and conclusions in the abstract and discussion

Was serum ferritin deficiency analyzed as a potential factor associated to h. pylori, similar to folate deficiency, or only the association of the actual values of serum ferritin were associated to h pylori? A similar approach should be used for folate and serum ferritin, and if not, it should be justified.

The sensitivity and specificity of the QuickVue should be reported in the paper, and it’s low specificity addressed. Similarly, Se/Sp for ELISA assays used should be reported in the methods, to be able to understand how these characteristics impact the results in the different age groups.

Where underage adolescents asked for consent to participate?

Adjusting by province may result in over adjusting and take away the effect of many variables such as differences in nutritional deficits between groups.

Using PCA to determine a wealth index is an not sufficient description to address what that index implies and how can it be interpreted, as it does not provide an standardize well known measure but instead it results in a data-specific index. Additionally, dichotomization of a data-rich summary probably reduces excessively the diversity and variance of the wealth index. Multiple, well-defined categories will probably would be a better fit for this data, particularly because it is trying to capture the diversity within a whole country.

Analysis of numerical covariates assuming linear relationships can lead to model misspecification and the shape of the actual association should be assessed before deciding on a linear or other type of fit, making sure that the model specification responds to the data

No rationale is presented for analyzing boys separate from girls

P values are used to define significance, but the actual values are not presented and without them is hard to assess the significance level

Reviewer #2: (No Response)

**Results**

-Does the analysis presented match the analysis plan?

-Are the results clearly and completely presented?

-Are the figures (Tables, Images) of sufficient quality for clarity?

Reviewer #1: (No Response)

Reviewer #2: (No Response)

**Conclusions**

-Are the conclusions supported by the data presented?

-Are the limitations of analysis clearly described?

-Do the authors discuss how these data can be helpful to advance our understanding of the topic under study?

-Is public health relevance addressed?

Reviewer #1: The discussion continues to be unfocused, particularly without attention to the nutritional factors that were supposed to be the main hypothesis

No attention is given to the broader age differences and trends between the three study groups, the role of tests used and the difference that age implies to the epidemiology of h pylori, as age trends seem to be present in children and adolescents, but not in adults. 

The limitations of the study are only listed and not really discussed or addressed as they may have important challenges to the conclusions and validity of the results

Reviewer #2: (No Response)

**Editorial and Data Presentation Modifications?**

Reviewer #1: English writing, only in the abstract: under five children, adolescent and … / We studied prevalence, / data from Nepal Micronutrient Survey / while blood sample was used / 

No hypothesis listed in abstract. 

Review comments are submitted with the understanding that the authors will respond to each comment by making changes in the actual manuscript to address each comment. Responses in the letter are useful, but actual changes in the manuscript are necessary, as readers will probably have similar concerns as those of reviewers.

Reviewer #2: (No Response)

**Summary and General Comments**

Reviewer #1: The changes made in this revised version of the manuscript do not feel that are properly addressing the concerns of this reviewer, as changes feel mainly superficial. A proper assessment of the role of nutritional deficiencies in H pylori prevalence would require more in-depth attention and focus

Reviewer #2: (No Response)

PLOS authors have the option to publish the peer review history of their article (what does this mean?). If published, this will include your full peer review and any attached files.

Reviewer #1: No

Reviewer #2: No

Figure Files:

Data Requirements:

Reproducibility:

References

---

## [Editor Report · Decision Letter 3]

27 May 2021

Dear Dr Mehata,

We are pleased to inform you that your manuscript 'Prevalence and correlates of Helicobacter pylori infection among under-five children, adolescent and non-pregnant women in Nepal: Further analysis of Nepal national micronutrient status survey 2016' has been provisionally accepted for publication in PLOS Neglected Tropical Diseases.

Best regards,

Jeffrey H Withey

Guest Editor

Robert Reiner

Deputy Editor

---

## [Editor Report · Acceptance letter]

16 Jun 2021

Dear Dr Mehata,

We are delighted to inform you that your manuscript, "Prevalence and correlates of Helicobacter pylori infection among under-five children, adolescent and non-pregnant women in Nepal: Further analysis of Nepal national micronutrient status survey 2016," has been formally accepted for publication in PLOS Neglected Tropical Diseases.

Best regards,

Shaden Kamhawi

co-Editor-in-Chief

Paul Brindley

co-Editor-in-Chief
